# Clinical Implications of Noncoding Indels in the Surfactant-Encoding Genes in Lung Cancer

**DOI:** 10.3390/cancers11040552

**Published:** 2019-04-17

**Authors:** Takahiro Nakagomi, Yosuke Hirotsu, Taichiro Goto, Daichi Shikata, Yujiro Yokoyama, Rumi Higuchi, Sotaro Otake, Kenji Amemiya, Toshio Oyama, Hitoshi Mochizuki, Masao Omata

**Affiliations:** 1Lung Cancer and Respiratory Disease Center, Yamanashi Central Hospital, Yamanashi 400-8506, Japan; nakagomi.takahiro@gmail.com (T.N.); shikarupd@yahoo.co.jp (D.S.); dooogooodooo@me.com (Y.Y.); lumi.hgc.236@gmail.com (R.H.); sotaro.otake@gmail.com (S.O.); 2Department of Surgery, School of Medicine, Keio University, Tokyo 160-8582, Japan; 3Genome Analysis Center, Yamanashi Central Hospital, Yamanashi 400-8506, Japan; hirotsu-bdyu@ych.pref.yamanashi.jp (Y.H.); amemiya-bdcd@ych.pref.yamanashi.jp (K.A.); h-mochiduki2a@ych.pref.yamanashi.jp (H.M.); m-omata0901@ych.pref.yamanashi.jp (M.O.); 4Department of Pathology, Yamanashi Central Hospital, Yamanashi 400-8506, Japan; t-oyama@ych.pref.yamanashi.jp; 5Department of Gastroenterology, The University of Tokyo Hospital, Tokyo 113-8655, Japan

**Keywords:** indel, untranslated region, lung cancer, mutation, targeted sequencing, cancer of unknown origin

## Abstract

Lung cancer arises from the accumulation of genetic mutations, usually in exons. A recent study identified indel mutations in the noncoding region of surfactant-encoding genes in lung adenocarcinoma cases. In this study, we recruited 94 patients with 113 lung cancers (88 adenocarcinomas, 16 squamous cell carcinomas, and nine other histologies) who had undergone surgery in our department. A cancer panel was designed in-house for analyzing the noncoding regions, and targeted sequencing was performed. Indels in the noncoding region of surfactant-encoding genes were identified in 29/113 (25.7%) cases and represent the precise cell of origin for the lung cancer, irrespective of histological type and/or disease stage. In clinical practice, these indels may be used as clonal markers in patients with multiple cancers and to determine the origin of cancer of unknown primary site.

## 1. Introduction

Large-scale sequencing of human tumor samples has implicated unexpected pathways and mutational processes in carcinogenesis [1,2]. The growing power of whole-genome sequencing now enables the discovery of significantly altered loci in noncoding sequences. The 98% of the genome that does not code for proteins includes transcribed but untranslated exons of genes, introns, and noncoding regulatory genetic elements, some of which may harbor clinically important and targetable DNA alterations [3,4]. In 2017, Imielinski et al. reported the presence of noncoding insertions/deletions (indels) in certain cancer types, which is reportedly a prevalent and hitherto unrecognized mutational process linking cellular lineage and cancer [5].

In that study, Imielinski et al. analyzed whole-genome sequences of lung adenocarcinoma using a somatic burden test based on Gamma–Poisson regression for the analysis of both indel and single nucleotide variant somatic mutations [5]. They found recurrent somatic indel mutations in noncoding regions of surfactant protein genes, the major transcriptional product of type II pneumocytes in the lung [5]. Using statistical analysis of whole-genome sequences across a diverse collection of cancers, they determined that other tumor types harbor similarly prevalent hotspots of noncoding somatic indel mutations, targeting lineage-defining genes (i.e., *ALB*, *TG*, and *LIPF*) [5]. These highly expressed genes define cell types that play essential biosynthetic roles in the physiology of their respective organs and constitute the precise cell of origin for the respective cancers.

The frequent indel mutation of this gene class is a previously undescribed feature of cancer genomes and quite surprising; however, it has yet to be elucidated whether these indels are a hallmark of carcinomas arising from secretory epithelial cell types or a more general phenomenon, and whether these indels are harbored by lung cancers other than adenocarcinoma. Furthermore, it remains to be determined at which stage of tumor phylogeny these noncoding indel mutations occur, how they are involved in oncogenesis, what effects they have on biological behavior, and what their clinical implications are.

As a rule, in whole-genome sequencing analysis the identification of spurious hotspots is quite broad, while the number of samples is limited compared with whole-exome capture data. Accordingly, based on the results of the above-mentioned study [5], we designed a custom panel for targeting the noncoding regions of six genes (*SFTPA*, *SFTPB*, *SFTPC*, *ALB*, *TG*, and *LIPF*). In our method, samples from surgical cases could be examined in our own laboratory, and the abundant clinical data in our facility could be analyzed for correlation. Namely, we investigated the correlation between these indels and the clinical, pathological and/or genomic findings, and attempted to characterize the cancers harboring these indels. In addition, we explored the possibility of a clinical application of these indels from the viewpoint of their clinical value as clonal markers and in the identification of the primary focus in cancers of unknown origin.

## 2. Results

### 2.1. Patient Characteristics

Ninety-four patients with lung cancer who had undergone surgery between July 2014 and March 2018 in our department were enrolled in this study in an unbiased manner. These subjects comprised 64 men and 30 women ranging in age from 49 to 90 years (mean 70.7 ± 12.5 years). Nineteen patients had double primary. Therefore, 113 cancers in total were analyzed. Histologically, they included 88 cases of adenocarcinoma, 16 cases of squamous cell carcinoma, four cases of pleomorphic carcinoma, four cases of small-cell carcinoma, and one case of large-cell carcinoma.

### 2.2. Detection of Noncoding Indels

A cancer panel was designed in-house based on data published previously to analyze the noncoding regions of six genes—surfactant protein A1, B, and C, as well as albumin, lipase, and thyroglobulin. Indels in the noncoding region of surfactant-encoding genes were identified in 29/113 (25.7%) cases (Appendix A). The detection rate of the indels in the three-surfactant protein (*SFTP*) genes was 6/113 (5.3%) in *SFTPA1*, 23/113 (20.4%) in *SFTPB*, and 2/113 (1.8%) in *SFTPC*. The co-occurrence of noncoding indels in *SFTPA* and *SFTPB* was found in 2 patients (Case 1 and 25 in Appendix A). These results were similar to those previously reported by Imielinski et al. (i.e., 6.0% in *SFTPA1*, 12.7% in *SFTPB*, and 5.1% in *SFTPC*) [5]. Allele fraction of detected noncoding indels was on average 34.4 ± 17.6%, relatively higher than that of the frequent mutations detected in the coding regions in lung cancer. In lung cancer specimens, no noncoding indels were detected in the albumin, lipase, or thyroglobulin genes, which indicated that the noncoding indels among the six genes are organ-specific, as demonstrated by a previous study [5].

As a confirmatory experiment, some of the detected noncoding indels, which exhibited high allele frequency, were validated through Sanger sequencing (Appendix A).

### 2.3. Patient Characteristics with or without Noncoding Indels

Lung cancers with and without noncoding indels were compared in terms of patient and tumor characteristics (Table 1). There were no significant differences in age, sex, performance status, smoking status, tumor size and location, CT findings, surgical procedure, pathological vessel and lymphatic invasion, or pathological stage.

### 2.4. Surgical Outcome with or without Noncoding Indels

There was no significant difference in either postoperative disease-free survival or overall survival between patients with lung cancers harboring noncoding indels and those not harboring noncoding indels (Figure 1).

### 2.5. Somatic Driver Mutations with or without Noncoding Indels

Targeted sequencing was performed using the cancer panel designed in-house to analyze the exon regions of 53 lung cancer-related genes [6,7]. When the frequencies of the affected pathways were compared between adenocarcinomas and squamous cell carcinomas, RAS (rat sarcoma viral oncogene homolog) pathway was found to be affected significantly more frequently in adenocarcinomas, while the chromatin remodeling, epigenetic, transcription, and TP53 pathways were affected significantly more frequently in squamous cell carcinomas (Appendix A). Meanwhile, there was no significant difference in the frequency of major driver mutations such as those of *KRAS*, *TP53* and *EGFR* in relation to the presence or absence of noncoding indels (*p* = 0.93, 0.92, and 0.43, respectively) (Figure 2).

### 2.6. The Presence of Noncoding Indels According to Histology 

Based on the histological classification, 23 (26.1%) of 88 adenocarcinomas, four (25.0%) of 16 squamous cell carcinomas, one (25.0%) of four pleomorphic carcinomas, and one (25.0%) of four small-cell carcinomas harbored noncoding indel mutations (Figure 3). Interestingly, noncoding indel mutations were detected in poorly differentiated carcinomas such as squamous cell and pleomorphic carcinomas at frequencies similar to those in adenocarcinomas. 

### 2.7. Noncoding Indels Arising in Various Histological Cancer Types

#### Case I

A 72-year-old man underwent left upper division segmentectomy because a ground-glass neoplasm was found in the left upper lobe (Figure 4A). The lesion was pathologically diagnosed as adenocarcinoma in situ (AIS), TisN0M0, stage 0 (Figure 4B). AIS is a well-differentiated adenocarcinoma, and the survival rate after resection is reportedly 100% [8]. Three indels in the noncoding region of the *SFTPB* gene were detected (Table 2). The detection of noncoding indels in a case of such early, well-differentiated lung cancer suggested that noncoding indel mutations occur in the early phase of tumor evolution.

#### Case II

A 79-year-old woman was found to have a massive tumor measuring 90 mm in diameter in the left lower lobe and underwent left lower lobectomy combined with resection of the thoracic wall and diaphragm (Figure 4C). Histologically, the lesion was diagnosed as squamous cell carcinoma, pT4N0M0, stage IIIB (Figure 4D). An indel in the noncoding region of the *SFTPB* gene was detected (Table 2). 

Thus, noncoding indels were detected in lesions ranging from early-stage lung cancer, as in case I, to massive and poorly differentiated cancer, as in this case.

#### Case III

A 78-year-old woman was found to have a massive tumor measuring 78 mm in diameter in the left upper lobe and underwent left upper lobectomy and chest wall resection. In this case, with pleomorphic carcinoma, histopathologically, the cancer was composed of papillary adenocarcinoma and sarcomatoid cancer, pT3N0M0, stage IIB (Figure 5A–C).

Whole-exome sequencing analysis of each component was performed, and phylogenetic analysis showed that the separate components shared several common mutations and that the sarcomatous component ramified from the carcinomatous component in the early phase of the evolutionary process and accumulated several mutations that were different from those of the carcinomatous component (Figure 5D). In terms of the noncoding indel mutations, the same noncoding indel was detected in both the adenocarcinoma and sarcomatoid cancer components (Table 2), and phylogenetic analysis revealed that this indel mutation occurred in the common trunk in the early phase of tumor evolution (Figure 5E). 

### 2.8. Noncoding Indels as a Clonal Markers for Multiple Lung Cancers

There are no specific clinical or radiological features that can be used to dependably distinguish multiple primary cancers from intrapulmonary metastases, and the differential diagnosis is at times perplexing in the clinical setting. As both a more precise and clinically applicable method for the diagnosis of multiple lung cancers, we performed mutation analysis of noncoding indels and evaluated the possibility of identifying the clonality of individual lung cancers using their mutations as a diagnostic marker.

Importantly, detection of noncoding indels in the *SFTP* gene was helpful in determining whether either of the tumors were primary or metastatic in nature, as shown in Cases IV–XIV.

#### Cases IV–XII

In cases IV–X, one of the two tumors harbored noncoding indels, whereas the other did not (Figure 6A–G). In cases XI and XII, *SFTPB* mutations were identified in both tumors, but they differed in nucleotide position and variance (Figure 6H,I). Thus, individual tumor in these patients did not share the same noncoding indel. In addition, the mutation profiles in the coding region also differed completely between the two tumors (Figure 6L–T). Such clear differences in mutation patterns supported the judgment of double primary cancer. 

#### Cases XIII and XIV

Targeted deep sequencing revealed that the noncoding indel mutation in 2 tumors in each patient involved *SFTP* (Figure 6J,K). Furthermore, the nucleotide position and variance in the mutation were entirely consistent (Table 2). Likewise, two lung cancers in each patient were characterized by identical gene mutations in the coding region (Figure 6U,V). Thus, a diagnosis of solitary, intrapulmonary metastasis was made in these cases.

### 2.9. Estimation of the Primary Focus in Cases of Carcinoma of Unknown Origin

Noncoding indels in the *SFTP* gene may indicate the precise cell of origin in lung cancer. Application of this finding to the clinical setting may provide an answer to the important clinical question as to whether mediastinal lymph node cancer is a form of lung cancer.

In addition to one surgical case, four cases of mediastinal lymph node cancer, although non-surgical cases, were also examined (Appendix A). Among five mediastinal lymph node cancers, noncoding indels were detected in two cases (Cases XV and XVI, Figure 7).

#### Case XV

Surgical resection of the lymph node revealed squamous cell carcinoma (Figure 7A,B). Immunohistochemically, tumor cells were negative for thyroid transcription factor-1 (TTF-1). Targeted sequencing was performed using a surgical specimen and it showed that the cancer harbored noncoding indels in the *SFTPA1* and *SFTPB* genes (Table 2). These indels represent an origin in the lung.

#### Case XVI

Endobronchial ultrasound-guided transbronchial needle aspiration (EBUS-TBNA) revealed that the tumor was adenocarcinoma (Figure 7C,D), but immunohistochemical stain for TTF-1 was negative. Using the EBUS specimen, targeted sequencing was performed and revealed that the cancer harbored noncoding indels in the *SFTPB* gene, leading to a diagnosis of mediastinal lung cancer (Table 2). 

In summary, *SFTP* noncoding indels were detected in two (Cases XV and XVI) of five cases of mediastinal lymph node cancer, and it was validated genomically that these mediastinal cancers originated from the lung (lung cancer).

### 2.10. SFTP Indels Near A or G Repeat Motifs 

The indel sites in our analysis were shown in Figure 8A. Furthermore, we examined the sequence features in 50-base neighborhood of highly expressed indels to identify the sequence contexts that might be associated with *SFTP* indel hotspots. Applying the Multiple Em for Motif Elicitation (MEME) algorithm (http://meme-suite.org/), we discovered significant enrichment of either an A repeat (E = 5.9 × 10^−10^) or G repeat (E = 5.9 × 10^−10^) motif in the vicinity of *SFTP* indels (Figure 8B).

## 3. Discussion

In 2017, Imielinski et al., using advanced analytical techniques, revealed that some lung adenocarcinomas harbor noncoding indel hotspots in surfactant protein genes (*SFTPA1*, *SFTPB*, and *SFTPC*) [5]. They obtained whole-genome sequencing reads mainly from the TCGA database and analyzed whole-genome sequences from 613 tumors representing 13 tumor types (79 lung adenocarcinomas, 50 lung squamous cell carcinomas, 64 cutaneous melanomas, 50 papillary thyroid carcinomas, 23 bladder cancers, 105 breast cancers, 32 glioblastomas, 16 head and neck squamous carcinomas, 54 hepatocellular carcinomas, 5 kidney cancers, 19 low-grade gliomas, 40 gastric cancers, and 76 prostate cancers), thus demonstrating a 25-fold enrichment of noncoding indels in *SFTP* gene (95% CI: (13.2, 47.4)) in lung adenocarcinoma versus other tumor types, even after correcting for sample-specific variations in indel density (*p* = 5.6 × 10^−23^, Wald test, logistic regression) [9,10,11]. In general, whole-genome sequencing leads to the enormous number of candidate regions; in addition, measurement is often difficult, and the number of cases with available data is limited. In our study, we intensively and efficiently searched for and analyzed the previously reported noncoding indels in our cases of surgically treated lung cancer, using a cancer panel designed in-house. The method with use of this cancer panel was advantageous in that it allowed comparison with abundant clinical data available in our facility, without limitation in the number of cases.

The indel hotspots are reported to target a special class of loci, encoding protein products that are manufactured in large quantities by a single cell type within a specific organ [5]. Moreover, the secretion of these lineage-defining proteins is a primary function of that cell type and vital to healthy organ and systems physiology. Surfactant proteins are surface-active agents produced and secreted by type II alveolar epithelial cells, representing major secretory proteins in the lung. *SFTP* is a lineage-defining gene in lung, and thus the noncoding indels in *SFTP* gene link cellular lineage and lung cancer [5]. 

Imielinski et al demonstrated that noncoding indels in the *SFTP* gene were detected exclusively in lung adenocarcinoma [5]. However, our study revealed that not only adenocarcinoma but also other types of poorly differentiated carcinomas such as squamous cell carcinoma and pleomorphic carcinoma in the lung, harbored noncoding indels. The frequency of noncoding indels in lung cancer had no association with the clinical findings such as stage and histology; indels therefore appear to be a general phenomenon that can occur at random in all lung cancers. Our study represents a follow-up experiment of the whole-genome sequencing study [5]; the following differences may exist between the two studies: (i) characteristics of the patients enrolled, (ii) depth of sequencing coverage, and (iii) DNA extraction methods. While Imielinski et al. used the data of whole-genome sequencing, we performed targeted deep sequencing; the mean sequencing coverage across targeted bases in our study was 1119X compared to 69X–121X in their study. Moreover, in our study, tumor cells were collected from formalin-fixed, paraffin-embedded (FFPE) tissues using a laser-capture microdissection system, and DNA was extracted after tumor cell collection and removal of normal tissues; this may have led to an increased allele fraction i.e., increased chance of detection of the targeted bases.

We previously reported that pleomorphic carcinoma composed of carcinomatous and sarcomatous components evolves from a common ancestral clone [12,13]. The phylogenetic analysis also showed that the sarcomatous component had ramified from the carcinomatous component in the early phase of the evolutionary process and accumulated several mutations that were different from those of the carcinomatous component [12]. In this study, noncoding indels were detected in both the carcinomatous and sarcomatous components in the pleomorphic carcinoma in case III. This finding indicated that this mutation occurs in the stage involving the common ancestral clone, i.e., in the early phase of tumor evolution [14]. This is also supported by the detection of noncoding indels in the early cancer lesion in case I with AIS.

In cases of multiple lung cancers, the clinical distinction between primary and metastatic tumors is sometimes difficult, making treatment selection challenging. Thus, we previously conducted lung cancer mutation analysis by targeted deep sequencing and found that mutations in individual lung cancers can serve as clonal markers, allowing identification of the clonality of individual tumors [15]. In this study, on the assumption that mutations in the noncoding region would also serve as clonal markers, we compared different tumors regarding the mutation pattern and found that noncoding indels in the *SFTP* gene can serve as clonal markers**.** In our experience, it is sometimes difficult to determine whether the tumor is primary or metastatic even when the data on mutations in the exon region is available; therefore, in such cases, noncoding indel data plays a supplementary role in diagnosis. Comparative approaches to the mutation patterns in the coding and noncoding regions, when combined, may help resolve the current dilemma of misdiagnosis in the clinical setting.

In general, cancers are composed of cell populations with distinct molecular and phenotypic features, a phenomenon termed intratumor heterogeneity [16,17]. Intratumor heterogeneity, associated with heterogeneous protein function, may foster tumor adaptation, phenotypic aggravation, and/or therapeutic failure through Darwinian selection [18,19]. In contrast, a pivotal driver mutation serves as the trigger of clonal expansion and is estimated to be retained homogenously within tumors of the same clone [20,21,22]. These events can be explained by “trunk and branch” mutation models, i.e., mutational events in early clonal progenitors are represented within the “trunk” of the tumor [16,17]. Such trunk somatic aberrations, present at the early stages of tumor development, are likely to occur ubiquitously at all sites of the disease. In contrast, later somatic events that occur following branched separation of subclones represent heterogeneous events. Such subclonal heterogeneity may be spatially separated between regions of the same tumor or its metastatic sites [16,17,18,19]. In this context, clonally dominant “trunk” aberrations are important clonal markers, and noncoding indels, with their allele fraction always being high at 11–92%, are considered “trunk” mutations. The detection of the same noncoding indel in different sites within the tumor in cases III also suggested that these indels are “trunk” mutations. Primary and metastatic tumors can be discriminated by determining whether such ubiquitous mutations are identical.

Cancer of unknown primary site (CUP) is the collective term for a group of cancers in which the anatomical site of origin remains unidentified after a metastatic focus had been found [23]. CUP is characterized by clinically unconfirmed primary malignancy, early occurrence of dissemination, rapid progression, and difficult prediction of the metastatic pattern [23,24]. The incidence of CUP is reported to be approximately 0.5–6.7% [25,26,27,28]. It occurs more frequently in men than in women and most frequently involves the lymph nodes and bones [28]. Histologically, most cases are adenocarcinomas [28]. CUP rarely occurs in mediastinal lymph nodes, accounting for only 1.0–1.5% of all CUP cases [27,28].

The lung is highly likely to be the primary site of metastatic mediastinal lymph node cancer. In approximately 40% of reported cancers of the hilar, mediastinal, and cervical lymph nodes, the site of primary malignancy was estimated to be the lung [28]. Taking into account the pathway of regional lymph flow in the lung, occult microcarcinoma can be considered to be present in the lung in cases of mediastinal lymph node CUP [29,30]. However, it is not rare for cancer of a distant organ, such as colorectal cancer or breast cancer, to metastasize to mediastinal lymph nodes [31,32,33,34,35], so identification of the primary focus has important implications for treatment selection. In particular, lung cancer is a prototype of precision oncology, and once a diagnosis of lung cancer is made, it is easier to develop effective treatment strategies using EGFR-TKI or anti-PD-1 antibodies as indicated by the tumor biomarkers. In fact, in cases XV and XVI in this study, immune checkpoint therapy using anti-PD-1 antibody achieved favorable control of the disease (Appendix A). Histopathological and immunohistochemical examinations are often helpful for the estimation of the primary origin, but almost never definitive. For instance, TTF-1 immunostaining has been conventionally used as an important marker in patients with unknown primary cancer. In the present study, TTF-1 immunostaining was negative in Cases XV and XVI, whereas the presence of noncoding indels revealed their origin in lung. Thus, it is assumed that noncoding indel may be a more accurate marker than TTF-1 immunostaining. Because *SFTP* is a lineage-defining gene for the lung, analysis of its noncoding indels can genomically and definitively demonstrate that CUP is derived from the lung. One limitation in terms of clinical application is that there is still a significant number of CUP of unknown origin owing to the low detection rate of *SFTP* noncoding indels in lung cancer. 

Much remains to be elucidated regarding the pathogenetic implications of noncoding indels. These mutations may be neutral and hence not affected by strong selection pressure. One possibility is that these mutations are not causally tied to the associated genetic lesions, but rather tag a particular evolutionary trajectory in tumorigenesis that is reflected in the transcriptional signature. The limitations of our study lie in the fact that it is based on data from a single institution and was performed retrospectively. Further molecular and translational studies, ideally on a multi-institutional basis, will be needed in the future to improve our understanding of the mutational processes and molecular pathogenesis of cancers and to make these findings more applicable to the clinical setting.

## 4. Methods

### 4.1. Study Population

The survey included 94 patients who had undergone surgery for lung cancer in our department between July 2014 and March 2018 and four patients not subjected to surgery but diagnosed with mediastinal lymph node cancer. All participants provided written informed consent prior to participation in the genetic research. The research was conducted in accordance with the Declaration of Helsinki, and the study was approved by the Institutional Review Board Committee of Yamanashi Central Hospital (Yamanashi, Japan). In total, 117 cancers were subjected to the mutation analysis, which included 75 patients with a single primary cancer, 19 patients with double primary lung cancer, and four patients with mediastinal lymph node cancer (non-surgical cases).

The medical records of the 94 consecutive patients with primary lung cancer who had undergone surgery at our hospital between July 2014 and March 2018 were retrospectively reviewed. The patients’ data were obtained from the cancer registry database of our institution. The information collected from the patients’ medical records included the preoperative characteristics, CT findings (tumor size and location), histopathological diagnosis, operative procedure, and follow-up data.

Histological typing was performed according to the World Health Organization’s classification (third edition) [36], and clinical staging was performed according to the International Union Against Cancer tumor–node–metastasis classification (eighth edition) [37,38].

### 4.2. Sample Preparation

Serial sections of FFPE tissues were stained with hematoxylin and eosin. Tumor cells were collected from FFPE tissues using an ArcturusXT laser-capture microdissection system (Thermo Fisher Scientific, Waltham, MA, USA). In case III, cancer cells were collected from histologically different components in each tumor by laser-capture microdissection. Panel sequencing was performed for each component, and the mutation profiles were compared.

DNA was extracted using a QIAamp DNA FFPE Tissue Kit (Qiagen, Hilden, Germany). FFPE DNA quality was evaluated using primers for the ribonuclease P (RNase P) locus [39]. A peripheral blood sample was collected from each patient just prior to surgery. The buffy coat was isolated following centrifugation, and DNA was extracted from the buffy coat using a QIAamp DNA Blood Mini Kit with a QIAcube system (Qiagen).

### 4.3. Targeted Deep Sequencing and Data Analysis

A previous report showed that indel mutations are frequently identified in surfactant protein genes (*SFTPA1*, *SFTPB*, and *SFTPC*) in lung adenocarcinoma, the albumin gene in liver cancer, gastric lipase gene in stomach carcinoma, and thyroglobulin gene in thyroid carcinoma [5]. Based on these results, we designed a custom panel to target the noncoding regions of these six genes. The target region covers 17.08 kilo base pairs, and 172 amplicons were included in this panel (Appendix A). Using this panel, we constructed a library and performed targeted sequencing, as described previously [40,41,42]. In brief, sequencing libraries were prepared using an Ion AmpliSeq Library Kit (Thermo Fisher Scientific). The barcode and adaptor were ligated, and library samples were purified using Agencourt AMPure XP reagent (Beckman Coulter, Brea, CA, USA). Library concentrations were determined using an Ion Library Quantitation Kit (Thermo Fisher Scientific). Emulsion PCR and chip loading were performed on the Ion Chef with the Ion Personal Genome Machine (PGM) Hi-Q View Chef Kit (Thermo Fisher Scientific), and sequencing was performed on the Ion PGM Sequencer (Thermo Fisher Scientific). 

The sequence data were processed using standard Ion Torrent Suite Software running on the Torrent Server. Raw signal data were analyzed using Torrent Suite. The pipeline included signal processing, base calling, quality score assignment, read alignment to the human genome 19 reference (hg19), quality control of mapping, and coverage analysis. Following data analysis, annotation of single nucleotide variants, insertions and deletions was performed using an Ion Reporter Server System (Thermo Fisher Scientific), and lymphocyte DNA from peripheral blood was used as a control to detect variants (tumor–normal pair analysis), as described previously [39,43,44].

### 4.4. Sanger Sequencing

PCR was performed using FFPE DNA as a template and primer pairs flanking the insertion and deletion sites. PCR products were purified using the ExoSAP-IT (Thermo Fisher Scientific) according to the manufacturer’s instructions. Sequencing was performed with BigDye Terminator v3.1. PCR products were purified with BigDye XTerminator Purification Kit (Thermo Fisher Scientific) and subsequently analyzed by the 3500 Genetic Analyzer (Thermo Fisher Scientific).

### 4.5. Whole-Exome Sequencing

Whole-exome sequencing and multiplex PCR were performed using buffy coat DNA and tumor DNA with an Ion AmpliSeq Exome RDY Kit (Thermo Fisher Scientific). The pooled PCR amplicons were treated with FuPa reagent to partially digest the primer sequences and phosphorylate the amplicons. The amplicons were ligated to adaptors with the diluted barcodes of the Ion Xpress Barcode adapters Kit (Thermo Fisher Scientific).

Adaptor-ligated amplicon libraries were purified using Agencourt AMPure XP reagent (Beckman Coulter). Each library was diluted, and the same amount of each library was pooled for a single sequence reaction. Emulsion PCR and chip loading were performed on the Ion Chef with the Ion PI Hi-Q Chef Kit (Thermo Fisher Scientific). Sequencing was performed using an Ion PI Hi-Q Sequencing Kit on the Ion Proton Sequencer (Thermo Fisher Scientific).

### 4.6. In Silico Analysis

For further inference of the phylogenies and estimation of the evolutionary distances, the neighbor-joining method was implemented to cluster the nonsilent mutations, and a phylogenetic tree was constructed [45]. The “ape” and “phangorn” (R version 3.2.3 in Linux) packages were used for these analyses. 

### 4.7. Motif Analysis

Genomic sequences (FASTA files) were extracted from ±50 base pairs upstream and downstream of the mutation sites using the UCSC genome browser. Sequence data were uploaded to MEME (http://meme-suite.org/tools/meme) to determine the enriched motifs near the mutation sites. 

### 4.8. Quantification and Statistical Analysis

Continuous variables are presented as the means ± SDs and were compared using unpaired Student’s *t*-test. One-way analysis of variance and the Tukey–Kramer multiple comparison test were used to detect significant differences between groups. Chi-square tests were used to compare the categorical data between groups. Recurrence-free survival was defined as the period from the day of operation to the day of recurrence or the day of final follow-up. Overall survival time was defined as the period from the day of operation to the day of death or that of the last follow-up evaluation. Survival was assessed using the Kaplan–Meier method, and comparisons among the survival curves were conducted using the log-rank test. All statistical analyses were performed using the JMP function in the SAS software package (SAS Institute, Inc., Cary, NC, USA). *P* values less than 0.05 in the two-tailed analyses were considered to denote statistical significance.

## 5. Conclusions

We identified indels in the noncoding region of surfactant-encoding genes in approximately 25% of lung adenocarcinoma, squamous cell carcinoma, pleomorphic carcinoma, and small-cell carcinoma cases. These indels indicate the precise cell of origin of lung cancer. In patients with multiple cancers, differences between the indels can be analyzed to determine whether the tumor is primary or metastatic. Furthermore, in patients with cancers of unknown primary sites, the lung can be identified as the primary site based on the presence of these indels. These findings strengthen the framework for implementation of indel detection in clinical practice as a method to enable identification of the origin of cancer and thus, improve treatment.

## Figures and Tables

**Figure 1 cancers-11-00552-f001:**
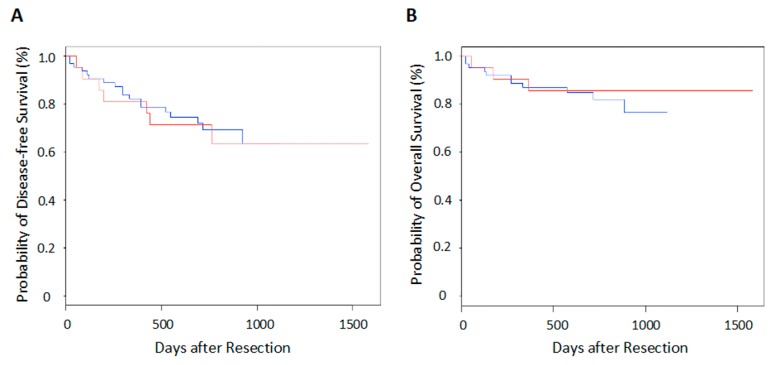
Survival after pulmonary resection in lung cancer patients with or without noncoding indels. (**A**) Disease-free survival, (**B**) Overall survival. The red lines denote the indel-positive group (*n* = 25) and the blue lines denote the indel-negative group (*n* = 69).

**Figure 2 cancers-11-00552-f002:**
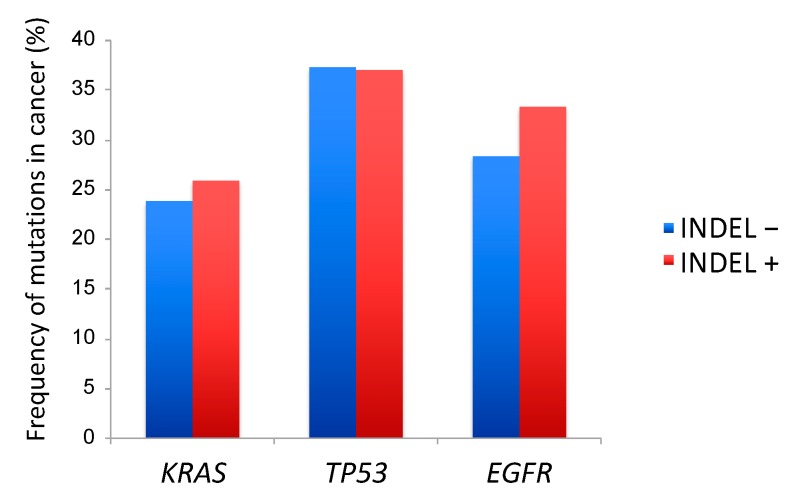
Mutational profile of lung cancer with and without noncoding indels. The prevalence of *KRAS*, *EGFR*, and *TP53* mutations was not significantly different between the cancers with and without noncoding indels.

**Figure 3 cancers-11-00552-f003:**
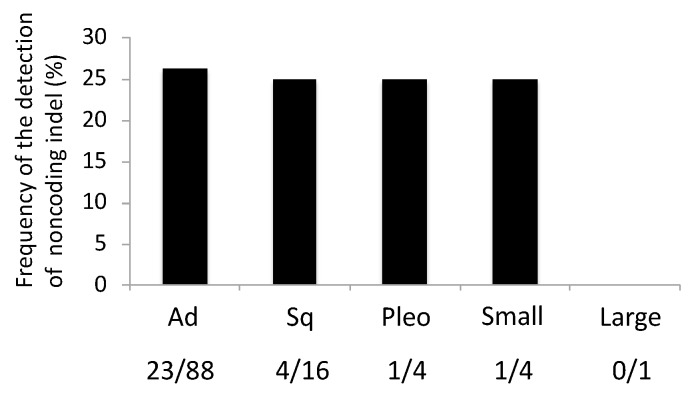
Mutation frequencies of noncoding indels in each histological type of lung cancer. Ad, adenocarcinoma; Sq, squamous cell carcinoma; Pleo, pleomorphic carcinoma; Small, small-cell carcinoma; Large, large-cell carcinoma.

**Figure 4 cancers-11-00552-f004:**
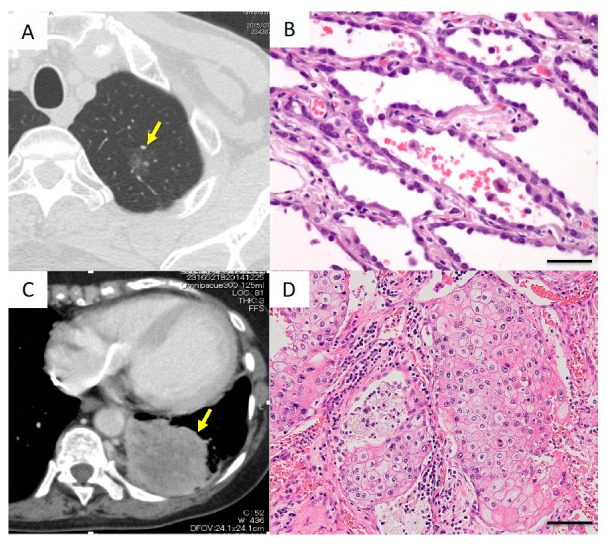
Radiological and histological findings in Cases I and II. (**A**,**B**) Case I. (**A**) Computed tomography (CT) image showing a ground-glass neoplasm in the apical portion of the left lung. The arrow indicates the lesion. (**B**) Histologically, the tumor was diagnosed as adenocarcinoma in situ. Scale bar, 20 μm. (**C**,**D**) Case II. (**C**) CT image showing a large mass lesion in the left lower lobe. The arrow indicates the lesion. (**D**) Histologically, the tumor was diagnosed as squamous cell carcinoma. Scale bar, 40 μm.

**Figure 5 cancers-11-00552-f005:**
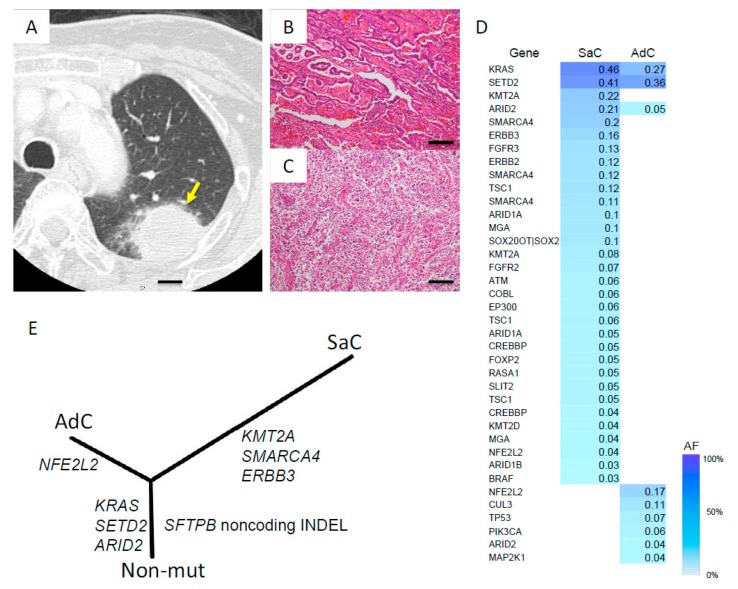
Radiological, histological, mutational, and phylogenetic findings in Case III. (**A**) CT image showing a large mass in the left upper lobe. The arrow indicates the lesion. (**B**,**C**) Histologically, the cancer was diagnosed as pleomorphic carcinoma, composed of adenocarcinoma and sarcomatous components (**B**: adenocarcinoma component, **C**: sarcomatoid component). Scale bars, 100 μm. (**D**) Heat map of the gene mutations in the coding region. This map shows the gene mutations of each component of the pleomorphic cancers. The column next to the heat map shows the mutated genes. SaC, sarcomatous component; AdC, adenocarcinoma component. AF allele fraction. (**E**) A phylogenetic tree was constructed using the whole-exome sequencing data and the bootstrap method. Noncoding indel mutation in the *SFTP* gene appeared to occur in the common trunk. Branch length indicates the evolutionary distance and is proportional to the number of nonsynonymous mutations. AdC, adenocarcinoma component; SaC, sarcomatous component; Non-mut, non-mutated cells.

**Figure 6 cancers-11-00552-f006:**
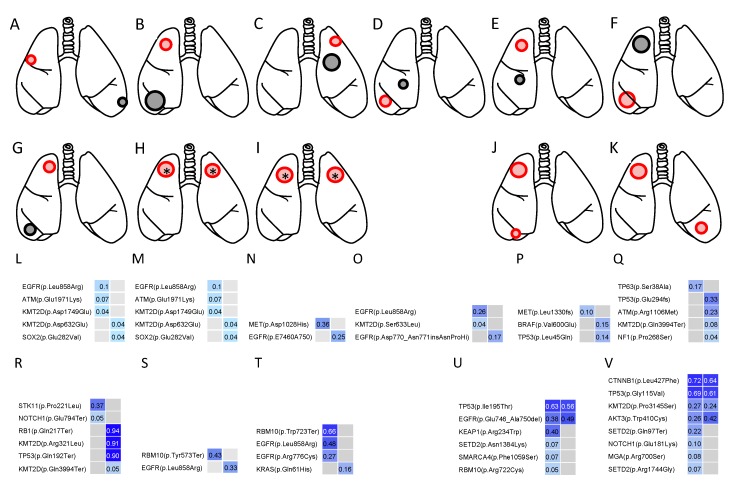
Cases IV–XIV with multiple lung cancers. Cases IV–XIV correspond to (**A**–**K**), respectively. (**A**–**I**) Schema of the two pulmonary lesions in case IV–XII. The red tumor in the schema harbored an *SFTP* indel, whereas the black tumor did not. Two lung cancers in each patient were characterized by different *SFTP* noncoding regions, and all the cases were judged as having double primary lung cancers. The asterisk (*) indicates different noncoding indels in the same gene (*SFTP*B). (**J**–**K**) Schema of the two pulmonary lesions in cases XIII and XIV. The two lung cancers were characterized by the same *SFTP* indel, suggesting the same clonality. (**L**–**V**) Heat map of the gene mutations in the coding region. Cases IV–XIV correspond to (**L**–**V**), respectively. (**L**–**T**) Two lung cancers in each patient were characterized by different gene mutations, and all the cases were judged as having double primary lung cancers. (**U**,**V**) The common driver mutations were shared by both lesions in each patient. AF, allele fraction.

**Figure 7 cancers-11-00552-f007:**
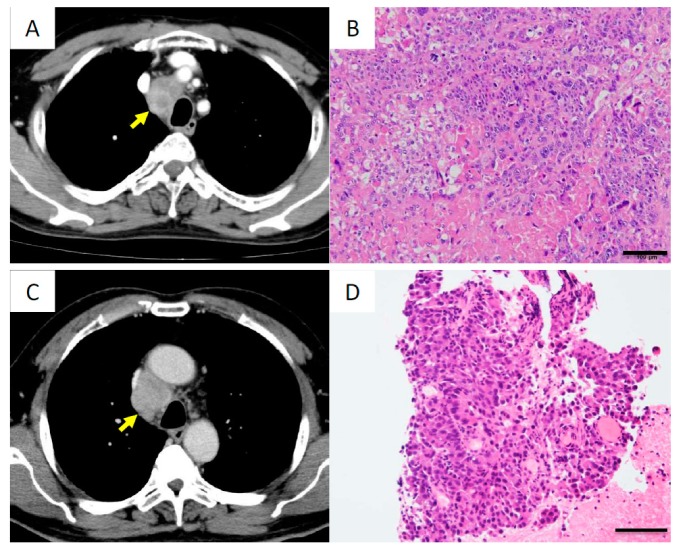
Cases XV and XVI with cancers of unknown primary site. Cases XV and XVI correspond to (**A**,**B**) and (**C**,**D**), respectively. (**A**) CT image showing a mediastinal mass in the upper paratracheal lesion, as indicated by the arrow. (**B**) Histologically, the tumor was diagnosed as squamous cell carcinoma. Scale bar, 100 μm. (**C**) CT image showing a mediastinal mass in the lower paratracheal lesion, as indicated by the arrow. (**D**) Histologically, the tumor was diagnosed as adenocarcinoma. Scale bar, 100 μm.

**Figure 8 cancers-11-00552-f008:**
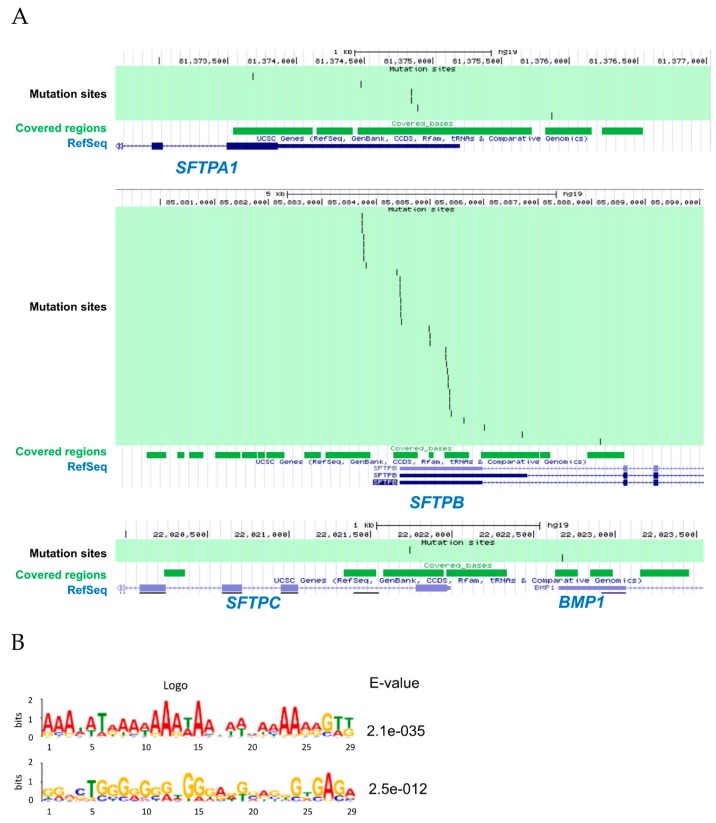
Molecular characterization of the noncoding indels in lung cancer. (**A**) Locus mapping of the noncoding indels. Black bar indicates the mutation site of the noncoding indel. (**B**) Motif analysis of *SFTP* indels. A or T repeat motifs were significantly enriched in the sequence neighborhood of *SFTP* indels, with sequences oriented to the (+) transcript strand of the associated gene. The size of the letters represents the frequency of the mutated nucleotides.

**Table 1 cancers-11-00552-t001:** Patient and cancer characteristics with and without noncoding indels.

Patients	Indel+ (*n* = 25)	Indel− (*n* = 69)	*p* Value
Age			0.41
median	69.6 ± 10.3	71.3 ± 13.3	
range	49–90	49–85	
Sex			0.34
male	15 (60.0%)	48 (69.6%)	
female	10 (40.0%)	21 (30.4%)	
Performance status			0.74
0	21 (84.0%)	56 (81.2%)	
1	4 (16.0%)	12 (17.4%)	
2–4	0 (0%)	1 (1.4%)	
Smoking status			0.66
never	18 (72.0%)	48 (69.6%)	
former/current	7 (28.0%)	21 (30.4%)	
Brinkman index			0.24
median	484.4 ± 135.3	629.6 ± 101.4	
range	0–1800	0–2520	
Pathological stage			0.81
0–IB	19 (76.0%)	52 (75.4%)	
IIA–IIB	3 (12.0%)	11 (15.9%)	
IIIA–IIIB	2 (8.0%)	4 (5.8%)	
IV	1 (4.0%)	2 (2.9%)	
Tumors	Indel+ (*n* = 29)	Indel− (*n* = 84)	
Location			0.54
central	2 (6.9%)	7 (8.3%)	
middle	10 (34.5%)	21 (25.0%)	
peripheral	17 (58.6%)	56 (66.7%)	
Computed tomography findings			0.46
solid	17 (58.6%)	50 (59.5%)	
part solid GGN	7 (24.1%)	19 (22.6%)	
GGN	5 (17.2%)	15 (17.9%)	
Surgery			0.69
wedge/segmentectomy	5 (17.2%)	14 (16.7%)	
lobectomy	23 (79.3%)	65 (77.4%)	
pneumonectomy	1 (3.4%)	5 (6.0%)	
Tumor size (mm)			0.44
median	25.6 ± 9.3	31.3 ± 8.4	
range	5–90	4–140	
Pathological vessel invasion			0.81
0	24 (82.8%)	72 (85.7%)	
1	4 (13.8%)	9 (10.7%)	
2	1 (3.4%)	3 (3.6%)	
Pathological lymphatic invasion			0.68
0	22 (75.9%)	66 (78.6%)	
1	6 (20.7%)	13 (15.5%)	
2	1 (3.4%)	5 (6.0%)	

Peripheral, central, and middle lung cancers were defined as cancers with primary lesions located in the outer, inner, or middle third of the lung field, respectively. GGN, ground-glass neoplasm.

**Table 2 cancers-11-00552-t002:** Mutation analysis of the noncoding indels in patients I–XVI.

Patient		Histology	Locus	Type	Ref	Mutation	Coverage	Gene	AF
I		Ad	chr2:85884386	deletion	TTTTTGGGATAA	T	1357	*SFTPB*	25%
			chr2:85885620	deletion	CT	C	1049	*SFTPB*	24%
			chr2:85888149	deletion	CAG	C	1472	*SFTPB*	27%
II		Sq	chr2:85886002	deletion	CT	C	1124	*SFTPB*	24%
III		Pleo, AD component	chr2:85883771	deletion	ATC	A	901	*SFTPB*	17%
		Pleo, SC component	chr2:85883771	deletion	ATC	A	1068	*SFTPB*	20%
**multiple lung cancers**									
IV	right	Ad	chr2:85886707	deletion	GCTT	G	926	*SFTPB*	27%
	left	Ad	-	-	-	-	-	-	-
V	right upper	Sq	chr2:85885333	deletion	AAT	A	353	*SFTPB*	52%
	right lower	Sq	-	-	-	-	-	-	-
VI	apical	Ad	chr2:85884386	deletion	TTTTTGGGATAA	T	1357	*SFTPB*	25%
			chr2:85885620	deletion	CT	C	1049	*SFTPB*	24%
			chr2:85888149	deletion	CAG	C	1472	*SFTPB*	27%
	hilar	Ad	chr2:85884998	deletion	CCTT	C	1336	*SFTPB*	35%
VII	right middle	Ad	-	-	-	-	-	*-*	-
	right lower	Ad	chr2:85884441	deletion	CTCAATC	CTC	1072	*SFTPB*	22%
VIII	right upper	Ad	chr2:85883742	deletion	ATTAT	A	527	*SFTPB*	19%
	right middle	Ad	-	-	-	-	-	*-*	-
IX	right	Sq	-	-	-	-	-	*-*	-
	left	Ad	chr2:85885334	deletion	ATATGACAAAAGTTATTTT	A	1281	*SFTPB*	31%
X	right lower	Ad	-	-	-	-	-	*-*	-
	right upper	Small	chr2:85884983	deletion	GAAT	G	1134	*SFTPB*	92%
			chr2:85885290	deletion	TTGTC	T	514	*SFTPB*	91%
XI	right	Ad	chr2:85883771	deletion	ATC	A	971	*SFTPB*	39%
	left	Ad	chr2:85885294	deletion	CTGTA	C	320	*SFTPB*	40%
XII	left	Ad	chr2:85883742	deletion	ATTAT	A	735	*SFTPB*	32%
			chr2:85884441	deletion	CTCAATC	CTC	599	*SFTPB*	26%
	right	Ad	chr2:85885286	deletion	CTCTT	C	684	*SFTPB*	13%
XIII	right upper	Ad	chr10:81374843	deletion	TCA	T	1016	*SFTPA1*	51%
	right lower	Ad	chr10:81374843	deletion	TCA	T	1743	*SFTPA1*	40%
XIV	right	Ad	chr2:85884451	deletion	CTT	C	1330	*SFTPB*	42%
			chr2:85885352	deletion	TATAA	T	1030	*SFTPB*	48%
	left	Ad	chr2:85884451	deletion	CTT	C	1244	*SFTPB*	34%
			chr2:85885352	deletion	TATAA	T	1357	*SFTPB*	35%
**mediastinal lung cancers**									
XV		Sq	chr10:81373682	insertion	T	TAGGCCTGACTG	258	*SFTPA1*	28%
			chr2:85885388	deletion	TAAG	T	594	*SFTPB*	18%
XVI *		Ad	chr2:85885352	deletion	TATAA	T	1123	*SFTPB*	41%

Ref, reference; Ad, adenocarcinoma; Sq, squamous cell carcinoma; Pleo, pleomorphic carcinoma; SC, sarcomatous; N.D., not determined; AF, allele fraction; *, non-surgical case.

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
