# Peer review of "Clinical Implications of Noncoding Indels in the Surfactant-Encoding Genes in Lung Cancer"

_cancers, 2019, doi:10.3390/cancers11040552_

Round 1
Reviewer 1 Report
The authors addressed most of the concerns in the initial submission. One minor comment is that I am still not sure how to read their mutation coding style in the mutation column of Table 2. Are insertions coded different from deletions?
Author Response
The authors addressed most of the concerns in the initial submission. One minor comment is that I am still not sure how to read their mutation coding style in the mutation column of Table 2. Are insertions coded different from deletions?
Response: We agree with the reviewer. Accordingly, Table 2 was revised, so that the readers could understand the pattern of mutations clearly: i) In the column “Type”, INDEL was reworded to either insertion or deletion. ii) In the column “Mutation”, only the single strand of DNA harboring the indel mutation was described. (Initially, we showed both normal and mutational strands of DNA, which might have been misleading.)
Reviewer 2 Report
The authors did not addressed all comments and should revise the manuscript prior to the publication, in particular regarding the statistical validity of the indels (validation) and overall diagnostic benefit with indels. The authors should screen other tumor types (e.g. GEO, TCGA) as independent control to strengthens the clinical role of the indels in lung cancer. Moreover, the study limitations should be extended.Author Response
The authors did not addressed all comments and should revise the manuscript prior to the publication, in particular regarding the statistical validity of the indels (validation) and overall diagnostic benefit with indels. The authors should screen other tumor types (e.g. GEO, TCGA) as independent control to strengthens the clinical role of the indels in lung cancer. Moreover, the study limitations should be extended.
Response:
Imielinski et al. was the first to report the finding that some lung adenocarcinomas harbor noncoding indels in surfactant protein genes(Cell, 2017, doi:10.1016/j.cell.2016.12.025.) In that study, utilizing mainly the TCGA database, they analyzed whole genome sequences from 613 tumors representing 13 tumor types. They obtained WGS reads for 79 lung adenocarcinomas, 50 lung squamous cell carcinomas, 64 cutaneous melanomas, 50 papillary thyroid carcinomas, 23 bladder cancers, 105 breast cancers, 32 glioblastomas,16 head and neck squamous carcinomas, 54 hepatocellular carcinomas, 5 kidney cancers, 19 low-grade gliomas, 40 gastric cancers, and 76 prostate cancers. They revealed that noncoding indels in SFTPgene were exclusively detected in lung adenocarcinomas, and demonstrated a 25-fold enrichment (95% CI: [13.2, 47.4]) in lung adenocarcinoma versus other tumor types, even after correcting for sample-specific variations in indel density (p = 5.6 x 10-23, Wald test, logistic regression).
We added these descriptions in the Discussion section.
As to the limitation of this study, we believe that it lies in the fact that the study is based on data from a single institution and was performed retrospectively.
We added these descriptions in the Discussion section.
Thank you very much for your thoughtful comments.
Reviewer 3 Report
Authors referred to all my comments and the manuscript has been improved. There is one mistake in the Table 1, row "smoking status": correct "066".
Author Response
Authors referred to all my comments and the manuscript has been improved. There is one mistake in the Table 1, row "smoking status": correct "066".
Response:
We agree with the reviewer. Accordingly, 066was corrected to 0.66.
This manuscript is a resubmission of an earlier submission. The following is a list of the peer review reports and author responses from that submission.
Round 1
Reviewer 1 Report
I read and revised the manuscript entitled “Clinical implications of noncoding indels in the 3 surfactant-encoding genes in lung cancer”.
In this study Takahiro et al investigated the role of non-coding Indels in the 3′ untranslated region of surfactant encoding genes and identified this genetic variant in 23/95 (24.2%) cases.
They also reported that these indels occur in the early phase of tumor evolution and indicate the precise cell of origin of lung cancer. Moreover, in patients with multiple cancers, differences between the indels can be analyzed to determine whether the tumor is primary or metastatic. They finally support the idea that these findings strengthen the framework for implementation of indel detection in clinical practice as a method to enable identification of the origin of cancer and thus, improve treatment.
The paper is well written and results are interesting although there are some issues that need to be evaluated and commented.
1- The number of samples is very small
2- Results reported in the “Detection of noncoding indels in each adenocarcinoma subtype” section are descriptive and there is a lack of a statistical analytical framework.
3- Indels have not been replicated and confirmed by a different method, moreover the sequence data were processed using the Ion PGM Sequencer and it’s well known that despite the PGM’s reported high accuracy in calling single nucleotide variations, it tends to generate many false positive or negatie calls in detecting insertions and deletions, which may hinder its utility for clinical genetic testing.
4- Results are very interesting but in general are descriptive and they lack a statistical solidity.
Author Response
1- The number of samples is very small
Response: We agree with the reviewer. The number of patients and samples is increased from 86 and 95 to 94 and 113 in total, as the reviewer suggested.
2- Results reported in the “Detection of noncoding indels in each adenocarcinoma subtype” section are descriptive and there is a lack of a statistical analytical framework.
Response: We agree with the reviewer and have deleted the section in the revised manuscript.
3- Indels have not been replicated and confirmed by a different method, moreover the sequence data were processed using the Ion PGM Sequencer and it’s well known that despite the PGM’s reported high accuracy in calling single nucleotide variations, it tends to generate many false positive or negative calls in detecting insertions and deletions, which may hinder its utility for clinical genetic testing.
Response: We agree with the reviewer. Accordingly, we have performed Sanger sequencing to confirm that the noncoding indels were not false positives but true indels.
4- Results are very interesting but in general are descriptive and they lack a statistical solidity.
Response: This study was designed to assess the presence of noncoding indels in surfactant genes in lung cancer. This study does not include a study intervention nor included a control group; therefore, we consider that this study is indeed a descriptive study.
Meanwhile, we have deleted section 2.7. “Detection of noncoding indels in each adenocarcinoma subtype”, because the statements made therein were very descriptive without substantial evidence, as suggested by the reviewer.
Reviewer 2 Report
Nakagomi et al. presented a manuscript that using target capture based deep sequencing to survey noncoding regions of 6 genes in lung cancer samples. The author claimed that this approach has clinical values. Although the detailed description of the result is presented in a relative clear fashion, it is largely descriptive and the results presented failed to convincingly support the conclusion.
Major comments:
1) The result showed that the presence of the noncoding indels in the surfactant-encoding genes is not associated with a long list of patient and tumor characteristics (molecular or clinical). This negative result largely suggests the lack of clinical value for the proposed approach. While the authors claimed that the approach can be used to reveal the relationship between two tumors from the same patient, this set of samples only accounts for 18 tumors/9 patients out of a total of 95 tumors/86 patients. Moreover, the approach failed to provides results for 4 out of 9 patients with multiple tumors. Therefore, even if we accept the authors’ claim at the face value, it works in 5/86 patients or 10/95 tumors surveyed, which is relatively inefficient.
2) While the presence of the same SNV/indel in two tumors from the same patient suggests that the two tumors were likely derived from a single ancestral clone, the existence of an indel in one but not the other tumor does not necessarily support that they are double primary tumors since it is possible (and expected) that additional mutations will be acquired in the metastasis tumor (or in primary tumor after the metastasis tumor seeded). The authors need to provide additional evidence to claim they are double primary cancers (case IV-VII).
Minor comments:
1) I am not sure how the indel was coded in Table 3. For example, in Case II, the ref allele is CT while the mutant allele is CT/C, does this mean one CT allele and one CC allele? The question is raised because the authors has coded the insertion explicitly in Case I chr2:85884386 (Ref: TTTTTGGGA, Mut: TTTTTGGGAT). If this is true, the claimed indel is actually a SNV instead.
2) The authors designed probes to capture non-coding regions of 6 genes, and sequenced around 1000X coverage. However, the lowest fraction of detected mutant allele is 17%, which can be safely detected using around 100X coverage, which may significantly reduce the sequencing cost. Please justify the design for a seemingly unnecessary high coverage.
3) Indel detection in NGS has been plagued with high false positives due to sequencing errors and/or alignment artifacts. Any validation experiment done for these detected indels using orthogonal methods? For example, for indels with a MAF arounf 0.5, a Sanger sequencing run will be able to test whether the reported indels are real or not.
4) The authors claimed that “These indels occur in the early phase of the tumor evolution…” (Line 413-414) without supporting evidence.
Author Response
Major comments:
1) The result showed that the presence of the noncoding indels in the surfactant-encoding genes is not associated with a long list of patient and tumor characteristics (molecular or clinical). This negative result largely suggests the lack of clinical value for the proposed approach. While the authors claimed that the approach can be used to reveal the relationship between two tumors from the same patient, this set of samples only accounts for 18 tumors/9 patients out of a total of 95 tumors/86 patients. Moreover, the approach failed to provide results for 4 out of 9 patients with multiple tumors. Therefore, even if we accept the authors’ claim at the face value, it works in 5/86 patients or 10/95 tumors surveyed, which is relatively inefficient.
Response:In the revised manuscript, we have increased the number of patients with multiple lung cancers from 9 to 19, and showed that 9 out of 19 patients with multiple lung cancers present double primary cancers based on the data of noncoding indels. In addition, we now show that 2 patients had intrapulmonary metastasis based on the congruence of noncoding indels.
Meanwhile, we are now able to determine whether the tumor was primary or metastatic in almost all patients with multiple lung cancers by combined analysis of differences in the exon region mutations and differences in the noncoding indel mutations. Evaluation of noncoding indel mutations can supplement the diagnosis even if the mutations in the exon regions are equivocal. This description has been added in the Discussion sections.
2) While the presence of the same SNV/indel in two tumors from the same patient suggests that the two tumors were likely derived from a single ancestral clone, the existence of an indel in one but not the other tumor does not necessarily support that they are double primary tumors since it is possible (and expected) that additional mutations will be acquired in the metastasis tumor (or in primary tumor after the metastasis tumor seeded). The authors need to provide additional evidence to claim they are double primary cancers (case IV-VII).
Response: Lung cancer panel covering all exons of 53 lung cancer-related genes was established in house, and deep sequencing was performed on each sample from patients with multiple lung cancers using this cancer panel. We compared the mutations in the exon regions in each tumor in patients with multiple lung cancers, and provided additional evidence indicating whether the tumor was primary or metastatic (Section 2.8. and Figure 6).
Minor comments:
1) I am not sure how the indel was coded in Table 3. For example, in Case II, the ref allele is CT while the mutant allele is CT/C, does this mean one CT allele and one CC allele? The question is raised because the authors has coded the insertion explicitly in Case I chr2:85884386 (Ref: TTTTTGGGA, Mut: TTTTTGGGAT). If this is true, the claimed indel is actually a SNV instead.
Response: We apologize for the oversight. In that table (originally created by Excel), there was a problem with conversion to Word, leading to wrong data. We have now presented the data correctly.
2) The authors designed probes to capture non-coding regions of 6 genes, and sequenced around 1000X coverage. However, the lowest fraction of detected mutant allele is 17%, which can be safely detected using around 100X coverage, which may significantly reduce the sequencing cost. Please justify the design for a seemingly unnecessary high coverage.
Response: In this study, tumor cells were, at first, selected from the FFPE samples of lung cancer by laser capture microdissection. Following this method, we found that the AFs of noncoding indels were generally high; however, this was unclear when we started this study, as there was no data regarding the AF. We believe that NGS with high coverage is an essential requirement of high quality research. Nonetheless, as suggested by the reviewer, this study can herein inform the readers that high coverage is unnecessary for similar analytical studies in future.
We added this finding in the Results and Discussion sections.
3) Indel detection in NGS has been plagued with high false positives due to sequencing errors and/or alignment artifacts. Any validation experiment done for these detected indels using orthogonal methods? For example, for indels with a MAF arounf 0.5, a Sanger sequencing run will be able to test whether the reported indels are real or not.
Response: For indels with a MAF around 0.5, a Sanger sequencing was performed, as the reviewer suggested. We confirmed that the detected indel mutations were true, as shown in Supplemental Figure S1. These confirmatory experiments were indeed essential for this study, and I appreciate the reviewer’s suggestion.
4) The authors claimed that “These indels occur in the early phase of the tumor evolution…” (Line 413-414) without supporting evidence.
Response: We agree with the reviewer. Accordingly, this sentence was deleted in the revised manuscript.
Reviewer 3 Report
The Authors of the present MS entitled “Clinical implications of noncoding indels in the surfactant-encoding genes in lung cancer” present a strategy for parallel screening of indel noncoding mutations of SFTP genes with the aim to identify the cancer origin with unknown primary site.Lung cancer arises from genetic mutations and a recent study identified indel noncoding mutations in SFTP genes. In this ms, targeted sequencing of indels in the 3′ UTR of SFTP genes identified 24.2% cases with the origin cell for the lung cancer.These mutationsmay be usedin clinical practice as clonal markers in patients to determine the cancer origin with unknown primary site.This method is based on before-published researches, already consolidated to analyze indel mutations in previous works.After careful review of the paper, I think that the topic of this study fits profile of “Cancers”. In my opinion, the presented paper is original, therefore acceptation has to be recommended after minor revisions.
I recommend Authors to perform English revisions of their paper to improve the flow of text.I suggest to the authors to use the automatic correction tool of word processor to carefully correct typing errors. Means ± SDs should be included in each figure where applied.
Author Response
I recommend Authors to perform English revisions of their paper to improve the flow of text.
Response:
Thank you for your suggestion. The manuscript has been checked by a professional English-editing service.
I suggest to the authors to use the automatic correction tool of word processor to carefully correct typing errors.
Response:
As suggested, the document was checked using the automatic correction tool of the word processor.
Means ± SDs should be included in each figure where applied.
Response: We have carefully checked this and included the necessary detail wherever applicable.
Reviewer 4 Report
Nakagomi et al. described the clinical implications of noncoding indels in the surfactant-encoding genes in lung cancer. The story is interesting, but has limitations:
- Indels in 23/95 cases is too low to conclude its clinical significance. I suggest, they should analyze sequencing data e.g. from GEO or TCGA to support the clinical significance of the indels with additional patient data (meta-analysis). Otherwise the 23 cases are too low.
- Moreover, further molecular characterization of the indels are neccessary.
- Did they diagnose the 10 reported cancer cases only based on the indels, or is it also possible with existing markers/tests?
- They should figure out the new insights/benefits of the indels compared to existing markers for diagnosis.
Author Response
1.- Indels in 23/95 cases is too low to conclude its clinical significance. I suggest, they should analyze sequencing data e.g. from GEO or TCGA to support the clinical significance of the indels with additional patient data (meta-analysis). Otherwise the 23 cases are too low.
Response: As the gene mutation data from GEO or TCGA database are not linked to any patient data, it is difficult to obtain such patient data. Therefore, in the revised manuscript, the number of patients and samples were increased from 86 and 95 to 94 and 113 in total, in response to the reviewer’s suggestion. As a result, we identified noncoding INDELs in 29/113, and have revised the clinical data in the manuscript accordingly.
2.- Moreover, further molecular characterization of the indels are necessary.
Response: We now provide a supplemental table including all data on noncoding INDELs. We showed that the AFs of identified noncoding INDELs were generally relatively high and considered that the indels were unlikely to be passenger mutations occurring during the course of tumor evolution.
Furthermore, we created a locus map, in which the noncoding indel mutation sites have been indicated (Figure 8A).
3.- Did they diagnose the 10 reported cancer cases only based on the indels, or is it also possible with existing markers/tests? They should figure out the new insights/benefits of the indels compared to existing markers for diagnosis.
Response:
We have added data regarding the mutations in the exon regions that were used to determine whether the tumor was primary or metastatic. In our experience, it is sometimes difficult to determine whether the tumor is primary or metastatic even when the data on mutations in the exon region is available; therefore, in such cases, noncoding INDEL data plays a supplementary role in diagnosis. This description was added to the Results and Discussion sections.
TTF-1 immunostaining has been conventionally used as an important marker in patients with unknown primary cancer. In the present study, TTF-1 immunostaining was negative in the two patients with cancer of unknown origin (Cases XV and XVI). We have noted that noncoding indel may be a more accurate marker than TTF-1 immunostaining in the Discussion section.
Reviewer 5 Report
This is a very interesting study, well designed and written. However, I have some minor comments:
1. Within the section 2.2 Authors write that indels in SFTP genes were detected in 23/95 cases (tumors) and that the distribution between particular SFTP genes was as follows: 6/95 in SFTPA1, 17/95 in SFTPB and 2/95 in SFTPC. There should be information which of them co-occurred. How these numbers correspond to the table 3?
2. The analysis of somatic point mutations was performed as well and the results for major driver mutations were presented in relation to the presence/absence of noncoding indels. However there is no information whether there were any differences in frequency of these mutations between particular lung cancer types that were analyzed. What was the distribution of somatic point mutations in particular lesions in Case III and other multiple lung cancers?
3. Section 2.7- Authors write that there was no significant difference between particular adenocarcinoma subtypes when it comes to the frequency of indels. However AIS and acinar subtypes were represented by only 4 cases. The small number of these groups should be commented since this is a limitation for such comparison.
4. Section 2.8- this section concerns noncoding indels in various histological cancer types and Authors present only two patients, one with AIS and one with squamous cell carcinoma. In turn, patient with the third type of analyzed lung carcinoma, the pleomorphic carcinoma, was presented in the next section. This should be written in another way.
5. Table 3 should contain noncoding indels in cases I-XI. However it contains cases I-X. This should be corrected. What does the word “case” mean? Patient or a tumor? Indels were detected in 22 patients (23 tumors) so why only cases I-X were presented within the Table 3?
6. When it comes to CT images there should be arrows pointing tumors.
7. Section 2.10- I understand that there was no case of multiple lung cancer in which particular lesions would share the same indel type? This should be commented.
8. The Authors write: “Nonetheless, because SFTP is a lineage-defining gene for the lung, analysis of its noncoding indels is able to genomically and definitively demonstrate that CUP is derived from the lung”. This is true only for positive results. However, there should be a comment that indels in SFTP genes are present in about 24% of lung tumors (according to Authors’ results) so still a significant number of such tumors remains of unknown origin.
9. Were these SFTP indels analyzed in normal noncancer lung cells?
Author Response
1. Within the section 2.2 Authors write that indels in SFTP genes were detected in 23/95 cases (tumors) and that the distribution between particular SFTP genes was as follows: 6/95 in SFTPA1, 17/95 in SFTPB and 2/95 in SFTPC. There should be information which of them co-occurred. How these numbers correspond to the table 3?
Response: As suggested, we have included the following in the Results section: The co-occurrence of noncoding INDELs in SFTPA and SFTPB was found in 2 patients.
In addition, we have provided Supplemental Table S1, which includes all data on noncoding indels.
2. The analysis of somatic point mutations was performed as well and the results for major driver mutations were presented in relation to the presence/absence of noncoding indels. However there is no information whether there were any differences in frequency of these mutations between particular lung cancer types that were analyzed. What was the distribution of somatic point mutations in particular lesions in Case III and other multiple lung cancers?
Response: As suggested, we have included information on the differences in exon mutations between adenocarcinoma and squamous cell carcinoma in Supplemental Figure S2. The somatic mutations in adenocarcinoma and sarcomatous components in pleomorphic carcinoma are shown in Figure 5D. We have also included information on the somatic mutations in the coding region in multiple lung cancer (Figure 6L-V).
3. Section 2.7- Authors write that there was no significant difference between particular adenocarcinoma subtypes when it comes to the frequency of indels. However AIS and acinar subtypes were represented by only 4 cases. The small number of these groups should be commented since this is a limitation for such comparison.
Response: We agree with the reviewer. Another reviewer also raised this point. Accordingly, we have deleted this analysis and the corresponding Table from the revised manuscript.
4. Section 2.8- this section concerns noncoding indels in various histological cancer types and Authors present only two patients, one with AIS and one with squamous cell carcinoma. In turn, patient with the third type of analyzed lung carcinoma, the pleomorphic carcinoma, was presented in the next section. This should be written in another way.
Response: We agree with the reviewer. In accordance with this suggestion, Cases I, II, and III have now been included in the same section.
5. Table 3 should contain noncoding indels in cases I-XI. However it contains cases I-X. This should be corrected. What does the word “case” mean? Patient or a tumor? Indels were detected in 22 patients (23 tumors) so why only cases I-X were presented within the Table 3?
Response: As Table 3 now shows data pertaining to cases I-XVI, the title of the table was changed accordingly. In the original manuscript, the word “case” indicated patient. In the revised manuscript, we have changed the word “case” to “patient” to avoid confusion. In addition, we have provided all data of noncoding INDELs in Supplemental Table S1.
6. When it comes to CT images there should be arrows pointing tumors.
Response: Thank you for your suggestion. We added arrows in Figure 4, 5, and 7 to point out the tumors.
7. Section 2.10- I understand that there was no case of multiple lung cancer in which particular lesions would share the same indel type? This should be commented.
Response: We agree with the reviewer. Individual tumors in cases IV-XII did not share the same noncoding indel. We added this description in section 2.8.
8. The Authors write: “Nonetheless, because SFTP is a lineage-defining gene for the lung, analysis of its noncoding indels is able to genomically and definitively demonstrate that CUP is derived from the lung”. This is true only for positive results. However, there should be a comment that indels in SFTP genes are present in about 24% of lung tumors (according to Authors’ results) so still a significant number of such tumors remains of unknown origin.
Response: We agree with the reviewer that there is still a significant number of CUP of unknown origin owing to the low detection rate of noncoding INDELs in lung cancer. We have added this description in the Discussion section.
9. Were these SFTP indels analyzed in normal noncancer lung cells?
Response: Unfortunately, we do not have data regarding the indels in non-cancer lung cells.